# The strength of association between psychological factors and clinical outcome in tendinopathy: A systematic review

Carl Stubbs[1], Sean Mc Auliffe[2]*, Adrian Mallows[3], Kieran O'sullivan[4], Terence Haines[5‡], Peter Malliaras[6]

1 Sunshine Coast Hospital Health Service, Monash University, Queensland, Australia, 2 Department of Physical Therapy and Rehabilitation Sciences, College of health Sciences, Qatar University, Doha, Qatar, 3 School of Health and Human Sciences, University of Essex, Colchester, United Kingdom, 4 Department of Allied Health, University of Limerick, Limerick, Ireland, 5 Department of Physiotherapy, School of Primary and Allied Health, Faculty of Medicine Nursing and Health Science, Monash University, Victoria, Australia, 6 School of Primary and Allied Health Care, Faculty of Medicine Nursing and Health Science, Monash University, Victoria, Australia

☯ These authors contributed equally to this work.
‡ These authors also contributed equally to this work.
* sean@qu.edu.qa

**Data Availability Statement:** All relevant data are within the manuscript and its Supporting information files.

## Abstract

### Objective

Tendinopathy is often a disabling, and persistent musculoskeletal disorder. Psychological factors appear to play a role in the perpetuation of symptoms and influence recovery in musculoskeletal pain. To date, the impact of psychological factors on clinical outcome in tendinopathy remains unclear. Therefore, the purpose of this systematic review was to investigate the strength of association between psychological factors and clinical outcome in tendinopathy.

### Methods

A systematic review of the literature and qualitative synthesis of published trials was conducted. Electronic searches of ovid MEDLINE, ovid EMBASE, PsychINFO, CINAHL and Cochrane Library was undertaken from their inception to June 2020. Eligibility criteria included RCT's and studies of observational design incorporating measurements of psychological factors and pain, disability and physical functional outcomes in people with tendinopathy. Risk of Bias was assessed by two authors using a modified version of the Newcastle Ottawa Scale. High or low certainty evidence was examined using the GRADE criteria.

### Results

Ten studies of observational design (6-cross sectional and 4 prospective studies), involving a sample of 719 participants with tendinopathy were included. Risk of bias for the included studies ranged from 12/21 to 21/21. Cross-sectional studies of low to very low level of certainty evidence revealed significant weak to moderate strength of association (r = 0.24 to

**Funding:** The author(s) received no specific funding for this work.

**Competing interests:** The authors have declared that no competing interests exist.

0.53) between psychological factors and clinical outcomes. Prospective baseline data of very low certainty evidence showed weak strength of association between psychological factors and clinical outcome. However, prospective studies were inconsistent in showing a predictive relationship between baseline psychological factors on long-term outcome. Cross sectional studies report similar strengths of association between psychological factors and clinical outcomes in tendinopathy to those found in other musculoskeletal conditions.

## Conclusion

The overall body of the evidence after applying the GRADE criteria was low to very low certainty evidence, due to risk of bias, imprecision and indirectness found across included studies. Future, high quality longitudinal cohort studies are required to investigate the predictive value of baseline psychological factors on long-term clinical outcome.

## Introduction

Tendinopathy, previously referred to as tendinitis or tendinosis, is a common musculoskeletal (MSK) condition characterised clinically by pain reported around the affected tendon with loading [1]. Tendinopathy affects both athletic and non-athletic populations alike. For example, Achilles tendinopathy is reported in up to 50% of runners before the age of 45 years [2]. In a Dutch general practice, Albers et al. [3] reported lower extremity tendinopathy prevalence rates of 11.8 per 1000 person-years, whilst prevalence rates for upper limb tendinopathies have been estimated between 1.3% to 21.0% [4–6].

Recommended care for tendinopathy includes progressive exercise interventions such as heavy-slow resistance (HSR), concentric and/or eccentric strengthening [7,8]. Success rates reported using such exercise programmes in Achilles tendinopathy have been shown to vary between 56% and 100%. [9,10] whilst moderate response rates of 41% have been reported for eccentric exercise in people with lateral elbow tendinopathy [11]. Given the often persistent and multifactorial nature of tendinopathy, similar to to other MSK disorders, treatments in tendinopathy may need to address the multiple factors that contribute to pain, dysfunction and disability experienced.

It has been suggested that psychological factors such as fear of reinjury, pain catastrophising, external locus of control and low self efficacy may negatively impact on clinical outcomes in common musculoskeletal disorders [12–15]. However, to date, the contribution of psychological factors to the pain, dysfunction and disability experienced in tendinopathy and the benefits of focussing upon theses factors as treatment targets remain uncertain. Recent qualitative studies have outlined the negative psychological impact of persistent Achilles tendinopathy, rotator cuff tendinopathy and greater trochanteric pain syndrome [16–19]. Likewise, a recent cross-sectional study reported greater levels of psychological distress and poorer quality of life among patients with more severe gluteal tendinopathy [20].

Previous systematic reviews [21,22] concluded that there was a conflicting evidence-base for the association of psychological factors and clinical outcome in people with tendinopathy. Several factors have been highlighted as potential explanations for the conflicting evidence base, such as, variance in population, heterogeneity of outcome measures and differing cognitive factors [21,22]. Despite the ambiguity, previous reviews suggest specific psychological variables may be associated with tendinopathy and suboptimal outcomes. Interestingly, no data

has yet been provided regarding the strength of association of psychological factors and outcome in people with tendinopathy and how they compare with other common MSK conditions. If psychological factors were shown to be strongly associated with pain, disability and physical functional outcomes, and if trials of management approaches that focus on these factors show them to be beneficial, it would provide stronger justification for targeting these factors in real-life management [21]. To address this, we aimed to identify and synthesise available evidence investigating the strength of association between psychological factors and clinical outcomes in tendinopathy.

## Methods

Protocol details were registered in the international prospective register of systematic reviews (PROSPERO—registration number CRD42019139757) and reported in accordance with the Preferred Reporting Items for Systematic Reviews and Meta-Analysis (PRISMA) [23].

### Eligibility criteria

Studies investigating a relationship between (i) psychological domains and (ii) clinical measures of pain and/or disability/function in tendinopathy were deemed suitable for inclusion. Randomised controlled trials (RCTs), quasi-experimental studies, cross-sectional uncontrolled trials, case-control, case series and cohort studies were eligible. Reviews, case studies, editorials, and studies with between group analysis were excluded.

### Participants

Studies which included adult participants labelled as having a tendinopathy of any duration, affecting the upper (rotator cuff, lateral elbow tendinopathy) or lower limb (plantar heel pain, Achilles tendinopathy, patellar tendinopathy, gluteal trochanteric pain syndrome), were included. In keeping with recent recommendations tendinopathy diagnosis could be based on clinical history and assessment, excluding other potential diagnosis, with or without imaging.

### Outcome measures

**Psychological outcomes.** Self-reported psychological instruments commonly reported in musculoskeletal literature measuring the following psychological factors were deemed suitable for inclusion in the review [24–26].

- **Emotional factors** including, but not limited to, depression, distress, anxiety, hypervigilance/somatisation, stress and anger.

- **Cognitive factors** including, but not limited to, maladaptive beliefs, fear, kinesiophobia, catastrophising, negative pain beliefs, self-efficacy.

- **Behavioural factors** including but not limited to avoidance, maladaptive (negative coping -styles.

**Clinical outcomes.** An a-priori decision was made to include the following clinical parameters relating to pain and disability as secondary outcomes: objective physical function, pain, self-reported disability, and any adverse effects.

- **Objective physical function**: Including but not limited to strength (isometric, isotonic, endurance), heel raise test, hop, balance and jumping tests.

- **Self-reported disability**: The data were extracted from the function scale highest on the following tendinopathy function scale hierarchy when trialists reported data for more than one scale.

  - Victorian Institute of Sports Assessment (VISA-P VISA-A, VISA-G).

  - Any other tendon specific disability scale (Shoulder Pan and Disability Index SPADI),

  - Global Rating of Change (GROC)

- **Pain**–Including, but not limited to, pain severity, overall pain, daily pain, pain with activity—including mean pain or change in mean overall pain measured via visual analogue scale (VAS), numerical rating scale (NRS) or categorical scale.

- **Adverse effects**: Number of participants experiencing an adverse effect (as defined by the trial authors)

## Timing of outcome assessment

Psychological and clinical outcomes were extracted and categorised into timelines. The following timelines were established, a priori, based upon previous literature [27,28]:

- Up to 4-weeks

- > 4-weeks and up to 3-months

- > 3-months

## Data sources and search

A systematic search of the following databases was performed by the lead author (CS): ovid MEDLINE, PsychINFO, Cochrane Library, ovid EMBASE and CINAHL plus. The search was originally performed in May 2019 and updated in June 2020. A sensitive search strategy using relevant search terms was developed from Medical Subject Headings (MeSH) and keywords were used (see Table 1). The complete search strategy for each database is shown in Appendix 1. Relevant grey literature was also searched via OpenGrey and ongoing trials via the National Institute of Health (clinicaltrials.gov). The electronic search was supplemented by hand searching of references lists from included articles.

## Selection of studies

Database screening was conducted by one author (CS) who screened titles and abstracts for potentially eligible trials based on predetermined criteria. Any potentially eligible studies along

**Table 1. Keywords used in the search strategy.**

| Main search terms |
|---|
| 1. Tendinitis or tendinopathy or epicondylitis or tennis elbow or shoulder impingement syndrome or rotator cuff injury or jumper's knee (MESH) |
| 2. Fear or depression or anxiety or catastrophizing or coping behaviour or avoidance behaviour or emotion or self-efficacy (MeSH) |
| 3. 1 and 2 combined |

with studies whose abstract did not provide enough information, were retrieved for full-text review and independently assessed by two authors (CS & AM) to determine eligibility. In cases of disagreement an agreement was made by consensus, and if required a third author was consulted (PM).

## Data extraction

Two authors (CS & PM) independently extracted data from each study to a standardised form. Disagreements were resolved through consensus, or through a third reviewer (SM) if required.

The following data were extracted from each study where available (see Table 2):

- Study characteristics (first author, year of publication, study design [e.g. case control], country, source of funding, sample size, trial registration [If registration number is reported]).

- Patient characteristics (mean age, mean duration of symptoms, sex, educational status, physical activity status, work status).

- Outcomes (psychological measures, function, disability, pain intensity, adverse events–as detailed above).

## Risk of bias and overall certainty

Two reviewers (CS & AM) independently assessed the risk of bias of each individual study using the Newcastle-Ottawa Scale (NOS) see Table 3. The NOS is a review tool for evaluating risk of bias in non-randomised studies [39,40]. An adapted version of the scale was selected to evaluate all studies because many were single cohort cross-sectional observational studies [41,42]. The tool consists of four domains of risk of bias assessment; (i) selection bias; (ii) performance bias; (iii) detection bias and; (iv) information bias. Seven items assess the four domains and each item is scored from zero (high risk) to three (low risk), with a total maximum score of 21 points. We then rated the overall risk of bias in each study as high (0–6), moderate (7–13), or low (14–21) in line with previous studies [39,40].

The overall certainty of the evidence was assessed by two teams of researchers (CS with either PM or SM) using the Grading of Recommendations Assessment, Development and Evaluation (GRADE) approach (see Table 4). The GRADE system classifies the certainty of evidence into four levels; high (further research is very unlikely to change our confidence); moderate (further research is likely to have an impact on our confidence); low (further research is very likely to have an important impact on our confidence); and very low (any estimate effect is very uncertain) [43].

The certainty of evidence is based upon five criteria (risk of bias, imprecision, inconsistency, indirectness and publication bias). As all eligible studies were observational designs, all studies started with a 'low quality' rating. Grading upwards was warranted if: 1) a large magnitude of effect existed; 2) there was a dose-response gradient, and 3) all plausible confounders and other biases increased our confidence of estimated effect [44].

## Data synthesis

Due to the heterogeneity of the included studies regarding body site, psychological domains, interventions, outcome measures and study design we were unable to pool data from individual studies to perform a meta-analysis. Qualitative synthesis was undertaken with studies grouped specific to each tendon site. Where possible, confidence intervals were calculated from Pearson's r effect sizes and unstandardised and standardised β values from individual

**Table 2. Characteristics of included studies.**

| First Author/ type of study | No of participants | Mean age +/- SD (years) | Clinical presentation | Duration of symptoms | BMI | Psychological factors outcome measures | Pain outcome measures | Disability Outcome measures | Physical Functional Outcome measures | Data Collection (follow up) | Statistical Results |
|---|---|---|---|---|---|---|---|---|---|---|---|
| Akyol et al. 2013 [29] Cross sectional study | 45 | 52.69±11.99 | Subacromial Impingement syndrome | 11.47±13.69 (SD) months | 28.93 ± 4.91 (SD) | Depression (BDI) QoL & Mental Health (SF-36) | (VAS 0–10) SPADI–pain VAS subscale | (SPADI) | SRS (IR) and (ER) velocity of 60°/sec and 180°/sec | T1 Baseline | Depression negative association with SRS 60°/s ERPT (r =−0.477, p = 0.001). Depression negative association with SRS 180°/s IRPT (r = -0.332, p = 0.026). Depression negative association with SRS 180°/s ERPT (r = -0.526, p = 0.001). Mental Health sub scale of SF-36 negative association with SRS 60°/s ERPT (r = 0.345, p = 0.020). Mental Health negative association with SRS 180°/s ERPT (r =−0.361, p = 0.015). Non-significant associations between Mental Health and SRS 60°/s & 180°/s IRPT. Depression was not significantly associated with SRS 60°/s IRPT. |
| Coombes et al. 2015 [30] Cohort prognostic study | 41 | 49.9 ± 7.4 | Lateral elbow tendinopathy | 25.7 ± 28.7 (SD) Weeks | - | Kinesiophobia–(TSK -17), 17–68, (where 0 is best) Depression/anxiety (HADS) 14 items rated on 4-point scales with a total score ranging from 0 to 42, Anxiety (0–21 with 0 being best); Depressive symptoms (0–21 with 0 being best). | Pain intensity (PRTEE) 5-items to assess pain scored on a 10-point likert scale (0-best) | Functional Disability (PRTEE)– 10-items on 10-point likert scale (0 is best). | - | (T1) at baseline (T2) at 2-months (T3) at 1-year | Psychological factors depression, anxiety or kinesiophobia (TSK, HADS) did not significantly influence either pain (PRTEE), disability (PRTEE) at 2 or 12-months HADS, per point at 52-weeks (β = -0.210; p = 0.087) |

*(Continued)*

**Table 2.** (Continued)

| First Author/ type of study | No of participants | Mean age +/- SD (years) | Clinical presentation | Duration of symptoms | BMI | Psychological factors outcome measures | Pain outcome measures | Disability Outcome measures | Physical Functional Outcome measures | Data Collection (follow up) | Statistical Results |
|---|---|---|---|---|---|---|---|---|---|---|---|
| Cotchett et al. 2015 [31] Cross sectional study | 84 | 56.07 ± 12.19 | Plantar heel pain | 13.6 ± 12.2 (SD)Months | 29.25 Range 20.9– 44.30 | Depression/anxiety/stress (DASS)- 0 to 21 for each subscale, where 0 is best and 42 is worst | FHSQ—Sub-SCALE, 0 is worst foot pain—100 is no foot pain | - | FHSQ—(0 is best foot function—100 is worst foot function) | (T1) baseline | Depression showed significantly negative association with foot function ($\beta$ = -0.28, p = 0.009). Stress showed significantly negative association with foot function ($\beta$ = -0.29, p = 0.006) Depression had a negative association with foot function in females ($\beta$ = -0.53; p<0.001). Depression was a significant predictor of foot pain in a model with females ($\beta$ = −0.41; p = 0.013) Stress was a significant predictor of foot pain in females in this model ($\beta$ = −0.36; p = 0.024) but not in males The association between stress and foot function was significant for females ($\beta$ = −0.50; p = 0.001) but not significant for males ($\beta$ = 0.01; p = 0.929) Anxiety was not a significant predictor of foot pain when added to a model with age, sex and BMI ($\beta$ = −0.04; p = 0.744); or when included in a model with stress ($\beta$ = −0.04; p = 0.783) or depression ($\beta$ = −0.02; p = 0.892). |

*(Continued)*

**Table 2.** (Continued)

| First Author/ type of study | No of participants | Mean age +/- SD (years) | Clinical presentation | Duration of symptoms | BMI | Psychological factors outcome measures | Pain outcome measures | Disability Outcome measures | Physical Functional Outcome measures | Data Collection (follow up) | Statistical Results |
|---|---|---|---|---|---|---|---|---|---|---|---|
| Cotchett et al. 2017 [32] Cross sectional study | 36 | 47.3 ± 13.2 | Plantar heel pain | 10.3 ± 12.8 (SD) Months | 24.6 ± 12.0 (SD) | Kinesiophobia (TSK-17) score 17–68, (where 0 is best) Pain catastrophising (PCS) score 0–52 (0 being best) | Pain intensity (FHSQ, SCALE, 0 is worst foot pain—100 is no foot pain); | - | FHSQ—(0 is best foot function—100 is worst foot function) First step pain (0–100 VAS, where 0 is best) | (T1) baseline | Univariate analysis Significant association between kinesiophobia and foot function ($r = -0.47$, $p = 0.005$) but not pain ($r = 0.01$, $p = 0.96$). Significant association between catastrophising and foot function ($r = -0.62$, $p = 0.000$) and pain ($r = 0.37$, $p = 0.025$). Multivariate analysis Kinesiophobia had a negative association with foot function ($R^2$ change 0.21, $F_{(1,31)} = 8.73$, $p = 0.006$) Catastrophising had a negative association with foot function $R^2$ of 0.39 f $_{(1,31)} = 21.142$, $p < 0.001$) Catastrophising explained 39% of the variability in foot function ($\beta = -0.65$, $P < 0.001$). Catastrophising and first step pain–$R^2$ of 0.18, F $_{(1,31)} = 7.918$, $P = 0.008$ catastrophising accounted for 18% of the variability in first step pain ($\beta = 0.44$, $P = 0.008$). |
| Engebretson et al. 2010 [33] Cross- sectional study | 200 | 49.8 ± 10.9 | Subacromial shoulder pain | Between 3-months and >24 months | | Emotional distress- anxiety and depression (HSCL-25) Consists of 25 items: Part I of the HSCL-25 has 10 items for anxiety symptoms; Part II has 15 items for depression symptoms. Scale rated 1–4 Overall score 25–100 with (25 being best) | Pain intensity during rest and activity (9-point Likert scale 1–9 with 1 being best) SPADI-Pain subscale | SPADI—0–100 where 0 is best. | - | (T1) baseline | Univariate regression Distress HSCL-25 significant association with SPADI total B = 13.9 (7.6 to 20.2) p<0.001 HSCL-25 significant association with SPADI pain subscale B = 13.6 (7.0 to 20.3) p<0.001 HSCL-25 significant association SPADI disability subscale B = 14.9 (7.8 to 21.9) p<0.001 Distress HSCL-25 significant association with SPADI total on multivariate regression B = 9.6 (3.0 to 16.2) P < .001 Distress HSCL-25 significant association on multivariate regression with SPADI pain subscale B = 10.1 (3.0 to 17.2) P = 0.006 Distress HSCL-25 significant association with SPADI on multivariate regression on disability subscale B = 10.4 (3.3 to 17.5) P < 0.001 |

(*Continued*)

**Table 2.** (Continued)

| First Author/ type of study | No of participants | Mean age +/- SD (years) | Clinical presentation | Duration of symptoms | BMI | Psychological factors outcome measures | Pain outcome measures | Disability Outcome measures | Physical Functional Outcome measures | Data Collection (follow up) | Statistical Results |
|---|---|---|---|---|---|---|---|---|---|---|---|
| Ferrer- Pena et al. 2019 [34] Cross—sectional study | 51 | 49.14 ± 8.693 | Greater trochanteric pain syndrome | 20.06 ± 28.658 months | 27.29 ± 5.62 (SD) | PCS score 0–52 (0 being best) TSK, 11 where 11 is best. Student t-test for independent samples was used to compare the outcomes between subjects who had lower levels of kinesiophobia with those with higher levels of kinesiophobia. No cut offs given. CPSS- 22 item questionnaire each item 10-point Likert scale anchored on the ends by 10 = very' uncertain and 100 = very certain (best = higher score) | Pain intensity (VAS 0–10 where 0 is best) | Spanish version of WOMAC—24 items divided into 3 subscales— scored on a scale of 0–4, Pain 0–20 Stiffness 0–8 Physical function −0–68 (0 = best) | Dynamic balance YBT. YBT composite score is calculated by summing the three reach directions (Forward, posteromedial posterolateral measured in cms) and normalizing to lower extremity length. YBT Forward established as criterion variable. | (T1)— baseline | Anterior YBT movement, negative associations between PCS and YBT-F $p < 0.05$. PCS subscale of helplessness revealed negative association— $p<0.05$ Positive associations were found in the pain subscale of the CPSS for anterior 0.432'*, postero-medial 0.313', and postero-lateral movements 0.309, other correlations for CPSS subscales not significant. Kinesiophobia was not significantly associated with three movements of YBT. Criterion variable (YBT-F) was predicted by VAS measures, the WOMAC total score, and the TSK-11 total score in multiple linear regression analysis. TSK-(B = 0.267, p = 0.02) Adjusted $R^2$ explaining 43.8% of variance |
| Harutaichun et al. 2019 [35] Prospective cohort study | 71 | 21.54 ± 1.14 | Plantar Fascitis | Between 2 and 10-weeks | 23.37 ± 3.81 (SD) | DASS (0 to 21 for each subscale, where 0 is best and 42 is worst) | Pain intensity (VAS)—No details regarding type of VAS (0–10 where 0-is best) | - | Ankle dorsiflexion angle subject in prone with knee extension measured with styandar goniometer Ankle plantarflexion strength—single leg heel raise—elevate heel every 2s in time with metronome. (norm 25 raises) Lateral step-down test from 15-cm step height. Score based on 5-criteria with scores range from 0–6 (where 0 = best) | (T1) Baseline | Depression and pain intensity positive association (r = 0.27, p = 0.03) Anxiety and pain intensity positive association (r = 0.38, p = 0.01) Stress and pain intensity positive association (r = 0.27, p = 0.03) Multiple linear regression analysis revealed participants with PF with reported higher levels of pain scored higher with anxiety–From the regression model there was a 0.13(cm) increase in pain intensity for each point of anxiety score. (B 0.41, p < 0.01) |

*(Continued)*

**Table 2.** (Continued)

| First Author/ type of study | No of participants | Mean age +/- SD (years) | Clinical presentation | Duration of symptoms | BMI | Psychological factors outcome measures | Pain outcome measures | Disability Outcome measures | Physical Functional Outcome measures | Data Collection (follow up) | Statistical Results |
|---|---|---|---|---|---|---|---|---|---|---|---|
| Kromer et al. 2014 [36] Cross-sectional longitudinal analysis | 90 | 51.8 (11.2) | Sub acromial shoulder pain | 38 (114) Median (IRQ) | - | FABQ, a 16-item questionnaire consists of 2-subscales (FABQ-PA) only one used. 4 items of scale are scored on a 7-point Likert scale (0-strongly disagree, 6-strongly agree (0 = best) PCS score 0-52 (0 being best) | Pain intensity (11-point VNRS) SPADI pain sub scale | SPADI contains 5 items assessing pain and 8 items assessing shoulder function. Only the SPADI-F in study. Each item is scored on a 100-mm visual analogue scale (0 = best) | - | (T1) Baseline (T2) 3-months | Baseline catastrophising and pain intensity positive association (r = 0.318, $p<0.01$). Baseline Fear avoidance and pain -nonsignificant association -0.061 $p>0.05$ Baseline fear avoidance and disability positive association (r = 0.237, $p<0.05$) Baseline catastrophising and disability positive association (r = 0.369, $p<0.01$) Fear avoidance beliefs (PA) positive association with baseline functional subscale (SPADI-F) on hierarchical regression analysis ($B = 0.287$, $p = 0.006$) Catastrophising non-significant association with baseline functional subscale (SPADI-F) on hierarchical regression analysis PCS and FABQ-PA non-significant association on SPADI-F at 3-months. |
| Maestroni et al. 2020 [37] Exploratory cross-sectional cohort study | 67 | 21.54 ± 1.14 | Rotator cuff related shoulder pain | 8-months (2-108) min, max. | (IQR) 23.5 (22.0–25.4) | Depression and anxiety. Two valid single questions were taken from the OEQ. 11-point scale with the endpoints 0 to 10 (where 0 is best) Fear avoidance—Single question asked from PAIRS 0 – "completely disagree" or 10 – "completely agree" (where 0 is best). | Measured using NRS, where 0 = no pain and 10 = pain as bad as it could be. | Disability was measured UEFI-20. 20 activities and the patient give a score to each one of them The highest score is 80 and the lowest is 0. A lower score indicates higher disability. | Maximal isometric strength of the shoulder internal and external rotator muscles. The isometric testing was performed in prone position with the arm supported in 90° of abduction and neutral rotation. Used electronic hand-held dynamometer | T1—Baseline | Significant positive association depression and pain intensity- regression analysis co-efficient 0.19 CI-(0.03 to 0.36) p = 0.020 Significant positive association anxiety and pain intensity Co-efficient 0.29 CI- (0.14 to 0.43) $p<0.001$ Significant negative association depression and disability Co-efficient −1.08 CI-(−1.84 to −0.32) p = 0.006 Significant negative association anxiety and disability – Co-efficient 1.08 CI-(−1.81 to −0.34) p = 0.005 Non -significant associations between fear avoidance beliefs and pain and disability Psychological factors not included in multivariable analysis. |

*(Continued)*

**Table 2.** (Continued)

| First Author/ type of study | No of participants | Mean age +/- SD (years) | Clinical presentation | Duration of symptoms | BMI | Psychological factors outcome measures | Pain outcome measures | Disability Outcome measures | Physical Functional Outcome measures | Data Collection (follow up) | Statistical Results |
|---|---|---|---|---|---|---|---|---|---|---|---|
| Silbernagel et al. 2010 [38] Case series | 34 | 51.6 ± 8.2 | Achilles Tendinopathy | 4.8 6 ± 0.2 years (SD) | - | TSK-SV-17 items, and the total score varies between 17 and 68. A high score indicates high degree of kinesiophobia. | - | VISA-A-S score ranges from 0 to 100; a lower score indicates more symptoms and greater limitation of physical activity | Jump tests—drop countermovement jump and hopping, Strength test—Concentric heel rise; eccentric-concentric heel rise (peak power in watts) Endurance Test—standing heel-rise with 10% of bodyweight added. (performed in joules). | (T1) 0-months (T2) 3-months (T3) 6-months (T4) 1-year (T5) 5-years | Kinesiophobia and heel-rise recovery significant negative correlation ($r = 0$–590, $p = 0.05$) TSK-SV(17) was no different between people who were asymptomatic, continued symptoms, or new symptoms at 5 years |

**BMI,** Body Mass Index; **BDI,** Becks Depression Inventory; **CPSS,** Chronic Pain Self-Efficacy Scale; **DASS,** Depression Anxiety Stress Scales; **ER,** External Rotation; **ERPT,** External Rotation Peak Torque; **FABQ,** the Fear-Avoidance Beliefs Questionnaire; **FABQ-PA,** Fear Avoidance Beliefs Questionnaire-Physical Activity; **FHSQ,** the Foot Health Status Questionnaire; **HADS,** the Hospital Anxiety and Depression Scale;; **HSCL,** the Hopkins Symptoms Checklist; **IR,** Internal Rotation **IRPT,** Internal Rotation Peak Torque; **NRS,** Number Rating Scale; **OMPSQ-SF,** Orebro Musculoskeletal Pain Screening Questionnaire-Short Form; **OEQ,** Outcome Evaluation Questionnaire; **PAIRS,** Patient and Impairment Relationship Scale; **PRTEE,** Patient Rated Tennis Elbow Evaluation; **PCS,** the Pain Catastrophising Scale; **SF-36,** the Short Form (36) Health Survey; **SSP,** Somatic trait anxiety Swedish University Scales of Personality; **SPADI,** the Shoulder Pain and Disability Index; **SPADI-F,** the Shoulder Pain and Disability Index; **SRS,** Shoulder Rotation Strength; **TSK,** the Tampa Scale for Kinesiophobia; **TSK-SV,** Swedish version of the Tampa Scale for Kinesiophobia; **UEFI,** Upper Extremity Functional Index; **VAS,** Visual Analogue Scale-Function; **VISA-A-S,** the Swedish Version of the Victorian Institute of Sports Assessment Achilles Questionnaire; **VNRS,** Verbal Numerical Rating Scale; **WOMAC,** Western Ontario McMaster Universities Osteoarthritis Index; **Y-BT,** Y-Balance Test.

**Table 3. Risk of bias (Newcastle Ottawa Scale- modified version).**

| First Author | Selection bias A | Performance Bias B | C | Detection Bias D | E | Information bias F | G | TOTAL | Level of bias |
|---|---|---|---|---|---|---|---|---|---|
| Akyol et al. 2013 [29] | 2 | 0 | 0 | 3 | 3 | 3 | 3 | 14/21 | Low |
| Coombes et al. 2015 [30] | 3 | 1 | 1 | 3 | 3 | 3 | 3 | 17/21 | Low |
| Cotchett et al. 2015 [31] | 3 | 3 | 2 | 3 | 3 | 3 | 0 | 16/21 | Low |
| Cotchett et al. 2017 [32] | 3 | 1 | 2 | 3 | 3 | 3 | 0 | 14/21 | Low |
| Engebretsen et al. 2010 [33] | 2 | 3 | 2 | 3 | 3 | 1 | 1 | 15/21 | Low |
| Ferrer-Pena et al. 2019 [34] | 2 | 3 | 0 | 2 | 3 | 3 | 3 | 14/21 | Low |
| Harutaichun et al. 2019 [35] | 3 | 3 | 3 | 3 | 3 | 3 | 3 | 21/21 | Low |
| Kromer et al. 2014 [36] | 3 | 3 | 2 | 3 | 3 | 3 | 0 | 16/21 | Low |
| Maestroni et al. 2020 [37] | 2 | 3 | 3 | 3 | 3 | 0 | 3 | 14/21 | Low |
| Silbernagel et al. 2010 [38] | 3 | 0 | 0 | 3 | 0 | 3 | 3 | 12/21 | Moderate |

A. Is the source population representative? **B**. Is the sample size adequate and is there sufficient power? **C**. Did the study adjust for confounders? **D**. Did the study use appropriate statistics to measure outcome of interest? **E**. Is there little missing data and was it handled appropriately? **F**. Are the methods of outcome measurements Explicitly stated and it is appropriate? **G**. Is there an objective assessment of outcomes?

studies. Pearson's r values between 0–0.2 very weak correlation; 0.2–0.5 weak correlation; 0.5–0.7 moderate correlation; and >0.7 strong correlation [45]. We calculated CI's for Pearson's r correlation coefficients using R Core team) [46,47].

## Results

### Study selection

4,155 studies were identified through searching electronic databases and 3 studies were identified from additional sources. We screened 3,562 titles and abstracts after removing duplicates and identified 44 for full-text review. A total of 10 observational studies (6-cross sectional and 4 prospective studies [3-cohort and 1-case series]) satisfied our eligibility criteria and were included in the final review. Fig 1 illustrates the selection process. We did not find any RCT's or quasi-experimental interventional studies in tendinopathy that had an intervention focussed on psychological factors.

**Table 4. Summary of findings and quality of evidence assessment (GRADE).**

| Outcome by tendon | No. of studies | Types of studies | No. of participants | Risk of bias | Inconsistency | Indirectness | Imprecision | Publication bias | Level of certainty |
|---|---|---|---|---|---|---|---|---|---|
| Rotator Cuff related Shoulder pain | 4-studies [29,33,36,37] | 3 x cross- sectional 1 x cross-sectional longitudinal | 402 | Serious (-1) | No | No | No | Undetected | low-level evidence |
| Plantar Heel Pain | 3-studies [31,32,35] | 2 x cross-sectional 1 x cohort prognostic study | 191 | Serious (-1) | No | No | Serious (-1) | Undetected | Very low-level evidence |
| Lateral Elbow Tendinopathy | 1- study [30] | 1 x cohort prognostic study | 41 | Serious (-1) | No | No | No | Undetected | low-level evidence |
| Greater Trochanteric Pain Syndrome | 1-study [34] | Cross sectional | 51 | Serious (-1) | No | Serious (-1) | Serious (-1) | Undetected | Very low-level evidence |
| Achilles Tendon | 1-study [38] | Case Series | 34 | Serious (-1) | No | No | Serious (-1) | Undetected | Very low-level evidence |

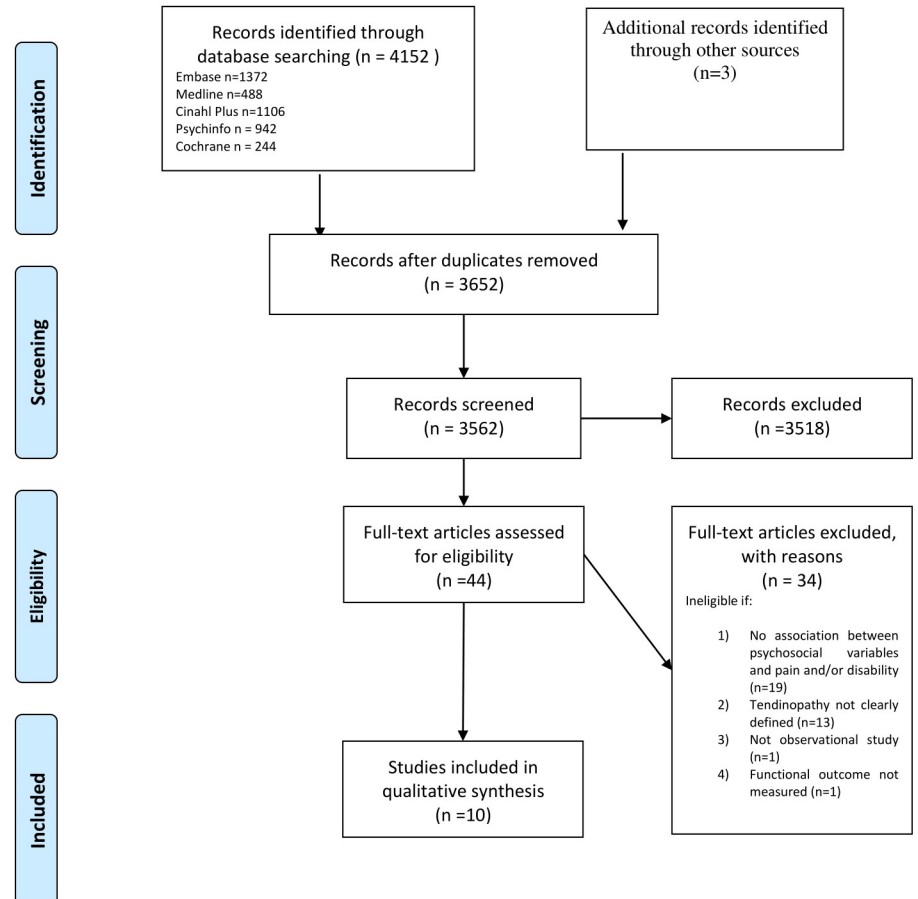

**Fig 1. PRISMA flow diagram.** Moher D, Liberati A, Tetzlaff J, Altman DG, The PRISMA Group (2009). *P*referred *R*eporting *I*tems of *S*ystematic Reviews and *A*nalyses: The PRISMA Statement. Plos Med 6(7): e1000097. doi:10.1371/journal.pmed1000097 **For more information, visit** www.prisma-statement.org.

## Characteristics of included studies

A total of 719 participants (49.1% females) with tendinopathy were included in the eligible studies. The mean age of participants across the included studies was 48 years of age with a mean duration of symptoms of 14.5 months. Four studies investigated rotator cuff related shoulder pain (subacromial impingement syndrome, subacromial shoulder pain, rotator cuff related shoulder pain) [29,33,36,37], three investigated plantar heel pain [31,32,35], one investigated lateral elbow tendinopathy [31], one investigated greater trochanteric pain syndrome [34] and the remaining study by Silbernagel et al. [38], investigated Achilles tendinopathy (see Table 2).

## Outcome measures

The outcome measures utilised across the 10 included studies are detailed in Table 2. For psychological factors, twelve-separate psychological outcome measures were used to assess a variety of emotional (5 outcome measures, Beck Depression Inventory (BDI), Hospital Anxiety and Depression Scale (HADS), Depression Anxiety Stress Scales (DASS), Hopkins Symptom Checklist-25 (HSCL-25), Outcome Evaluation Questionnaire (OEQ)) and cognitive

(7 outcome measures, Tampa Scale of Kinesiophobia-17 (TSK-17), Tampa Scale of Kinesio-phobia -11 (TSK-11), Tampa Scale of kinesiophobia Swedish Version -17 (TSK-SV-17), Patient Catastrophising Scale (PCS), Chronic Pain Self-Efficacy Scale (CPSS), Fear Avoidance Belief Questionnaire (FABQ), Patient and Impairment Relationship Scale (PAIRS)) domains. The most common emotional factors measured were depression (6 studies, 5 separate outcome measures) and anxiety (5 studies, 4 separate outcome measures). The most common cognitive factor measured was kinesiophobia (4 studies, 3 outcome measures), see Table 2.

For the secondary outcomes, 10 studies included 6 separate outcome measures for pain with pain intensity using the visual analogue scale being the most frequent (3 studies). Objective physical function was measured in 7 studies and utilising 10 separate outcome measures, including isometric and isokinetic strength, balance, jumping and hopping. Finally, 7 studies used 5 separate outcome measures for self-reported disability, all of which were region-specific. No studies reported adverse effects which is not surprising given none were trials.

## Risk of bias

The overall risk of bias for 9/10 studies was low whilst one study was considered to be at moderate risk of bias. Scores ranged from 12/21 to 21/21 (see Table 3). The most common sources of bias were; not adjusting for confounders, (81.8%), inadequate sample size and statistical power (54.5%) and a lack of objective outcome assessment (45.6%).

## Overall certainty of evidence

Due to heterogeneity regarding tendon sites, psychological variables and outcome measures investigated, we were unable to apply the GRADE criteria to measure the certainty of evidence for each individual patient outcome. Consequently, the GRADE criteria was used to establish the certainty of association as it was intended and this was also equivalent to each individual study. Overall, the GRADE criteria demonstrated low to very low levels of certainty (see Table 4). Limitations mostly related to risk of bias (80%), imprecision (40%) and indirectness (10%). Observational studies included in our review had a greater potential for risk of bias due to a lack of randomisation which increases the possibility of confounding, and selection bias [48]. When considering imprecision, we rated down the evidence quality in four of our studies due to the lack of reporting of 95% confidence intervals.

## Associations between psychological factors and clinical outcomes

The specific associations investigated for each study are detailed in Table 2. Some representative examples are provided for each tendinopathy/tendon region below.

## Rotator cuff tendinopathy

There was low certainty evidence from one study of prospective design [36] supporting a weak positive baseline association between catastrophizing and pain (r = 0.32, p<0.01) and disability (r = 0.37, p<0.01). Fear avoidance beliefs measured at baseline appeared to be significantly associated with baseline disability (r = 0.237, p<0.05) but not significant with disability change scores after 3-months.

There was low certainty evidence from three cross-sectional studies supporting a weak positive association between baseline psychological factors (including depression, anxiety and emotional distress) and pain [33,37], disability [33,37] and a negative association with physical function [29]. Thirty four percent (10/29) of the associations investigated were not significant, (refer to Table 2) with some associations differing between clinical outcomes. For example,

Kromer et al. [36] showed fear avoidance behaviour was weakly associated with baseline disability but not pain intensity. Furthermore, preliminary cross-sectional associations found between psychological factors and disability were not always evident when confounding variables were considered using multivariate regression models [33] or when measured at long-term follow up [36].

## Plantar heel pain

Very low-level certainty evidence from a prospective cohort study found a weak positive association between pain intensity and both depression, (r = 0.27 (95% CI = 0.02 to 0.45) p<0.03 and stress (r = 0.27 (95% CI = 0.04 to 0.47 p<0.03) among men with plantar heel pain [35]. Two cross sectional studies demonstrated a weak to moderate association between psychological factors (kinesiophobia, catastrophizing, depression and stress) and foot function and pain [31,32]. Forty seven percent (16/34) of associations investigated were not significant (refer to Table 2). Once again, the associations between clinical and psychological factors varied between studies. For example, cross sectional analysis revealed anxiety was not significantly associated with foot pain [31], whereas, a recent prospective cohort study showed anxiety to be the strongest predictor of pain intensity in people with plantar heel pain ($\beta$ = 0.41, $p$ = 0.01) [35].

## Lateral elbow tendinopathy

Low certainty evidence from one cohort prognostic study suggested psychological factors (depression and kinesiophobia) were not significantly associated with either pain or disability in patients with lateral elbow tendinopathy at 2 or 12-months [30].

## Greater trochanteric pain syndrome

Very low certainty evidence from one cross-sectional study investigated the relationship of psychological factors (including kinesiophobia, self-efficacy and catastrophizing) and objective physical function (dynamic balance) [34]. Those with higher catastrophising (helplessness component) were negatively associated with poor performance in components of the Y-balance Test (r = -0.304 (95% CI = -0.53 to—0.03) p<0.05) whilst greater self-efficacy (pain component) revealed a weak positive association with the anterior component of the Y-Balance Test (r = 0.432 (95% CI = 0.18 to 0.62) p<0.01). Multiple linear regression analysis revealed greater kinesiophobia, along with higher pain and poorer WOMAC total score, contributed to a poorer score on the anterior component of the Y-Balance Test.

## Achilles tendinopathy

There was very low certainty of evidence from a 5-year prospective case series by Silbernagel et al. [38] showing a significant moderate correlation between the level of kinesiophobia (fear of movement), and performance in a repeated standard heel raise functional test (r = -0.590 (95% CI = -0.80 to -0.2 p<0.005).

## Discussion

This systematic review investigated the relationship between psychological variables and clinical outcomes in tendinopathy. Ten observational studies were included, including six cross-sectional and four prospective studies. Among the cross-sectional studies, there was low to very low certainty evidence for an association between emotional (e.g. stress, depression) and cognitive (e.g. kinesiophobia, fear-avoidance) issues and greater self-reported pain and

disability as well as impaired physical function in people with tendinopathy. In addition, there was low to very low certainty evidence for an association between higher levels of self-efficacy and lower levels of pain intensity. In the context of our review the GRADE evaluation suggests we have low to very low certainty in the associations identified in this review due to factors such as risk of bias, imprecision and indirectness, hence we have very little confidence in the effect estimate [43]. Four prospective studies revealed a conflicting association between psychological factors and change in clinical outcomes. This is a key limitation of this literature, prospective studies are required to explore cause and effect relationships [49].

The strength of association in the cross-sectional data (r = 0.24 to 0.53) is comparable to other reviews among people with other musculoskeletal conditions [50–52]. In a review of 118 cross-sectional studies in low back pain, Kroska et al. [51] reported very weak to moderate positive associations between fear-avoidance and pain (r = 0.01 to 0.65). Whilst, Luque-Suarez et al. [52] included 50 cross sectional studies in their review and reported very weak to moderate strength of association between kinesiophobia and pain in people with an array of different chronic musculoskeletal pain conditions (r = 0.03 to 0.58). Taken together with findings of the current review, psychological factors and clinical outcomes appear to share a modest association for people with musculoskeletalal conditions. This suggests, that at least for some people with musculoskeletal conditions, psychological factors are associated with clinical outcome, and worthy of consideration. For example, some people may have low fear avoidance but high pain ratings, and the reverse may also be true. Identifying people for whom psychological factors relate to pain, disability and functional outcomes may be important for a more targeted management approach.

In contrast to cross-sectional studies, longitudinal studies can evaluate whether psychological factors predict disease outcome over time. Studies of prediction require prospectively collected longitudinal data where the outcome is not present at enrolment [53]. This temporal sequence is important in determining cause and effect [54]. The data from the four longitudinal studies were inconsistent for the predictive value of psychological factors on long term clinical outcome [30,35,36,38]. This may in part be explained by methodological heterogeneity including variability in tendon sites (4 sites), psychological factors measured (6 factors), clinical outcomes assessed (7 outcomes), and assessment periods (0–60 months).

It is also noteworthy that many people in the included trials had low baseline scores for the psychological constructs assessed. People with low baseline psychological scores may be less likely to display a relationship between their psychological status and clinical outcomes [55]. For example, baseline mean values for kinesiophobia (TSK) were reported as 23.7 points by Coombes et al. [30]; scores between 17–37 points are considered moderate risk of poor outcome [56,57]. In the same study, participants recorded mean scores of 7.6 out of a total of 42 on the Hospital Anxiety and Depression Score (HADS), on which a score of 7 points (or lower) is considered not anxious or depressed [58]. Similarly, median baseline catastrophising was reported as 9 points by Kromer et al. [36], whereas scores >30 points are considered clinically relevant [59]. As higher levels of psychological factors at baseline may predict poorer clinical outcomes [57,60], it appears this population may simply be underrepresented in the included studies; perhaps due to sampling bias or strict inclusion/exclusion criteria which often excludes participants with high levels of pain or bilateral pain symptoms [33,36,37].

Similarly to a prior review [21], we found substantial heterogeneity in relation to the range of psychological outcome measures across the included studies included in our review. For instance, depression and anxiety was measured in six studies, in which six different outcome measures were used. This lack of concensus restricts the synthesis and pooling of data for meta-analysis.

In accordance with other reviews [50,51,61,62] there was consistency found in instruments used to measure fear avoidance (FABQ and kinesiophobia (TSK). Whilst the contributing role of fear avoidance behaviours has been well established in the development of chronic lower back pain and other musculoskeletal disorders [51,63,64], historically, this relationship has not been adequately examined in tendinopathy studies. Hence, it is possible that people with tendinopathy may not experience the same magnitude of fear and subsequent avoidance as do people with other musculoskeletal conditions, thus, these instruments may not adequately capture the lived experience of people with this musculoskeletal condition.

## Strengths and weaknesses

Strengths of the review include pre-registering the protocol, a rigorous search strategy combined with using validated methods for evaluating risk of bias and the level of evidence. The review has been reported in accordance with PRISMA guidelines [23] and utilised two reviewers to screen and extract data. The main limitations relate to the literature. Most studies were cross-sectional with small samples, which are more prone to bias and confounding than prospective studies, and are unable to identify temporal relationships that may be causal. Second, high levels of heterogeneity prevented meta-analysis and limited comparisons between studies. Finally, detailed descriptive statistics were not always available, making the task of summarising these studies consistently challenging.

## Future directions

First, there is a clear need for robust longitudinal studies to investigate the predictive value of psychological factors among people with tendinopathy. Second, significant heterogeneity of psychological outcome measures exist in selected studies, resulting in inconsistency of reporting and thus preventing meta-analysis of the data; This lack of standardisation of outcome measures limits the usefulness of clinical trial evidence [65]. Within rheumatology, this problem has been addressed through the implementation of the Outcome Measures Rheumatology initiative (OMERACT), which has markedly improved outcome measurement for many rheumatologic conditions. This has been achieved by developing widely endorsed "core outcome measurement sets" (COMS) that are to be reported in all RCT and longitudinal studies [65,66]. Recently, the International Scientific Tendinopathy Symposium Consensus (ICON) group have identified nine health related core domains, one being psychological factors [67]. Future studies are required to establish which psychological instruments would be applicable to use in the chosen setting. However, first we need to determine the specific psychological constructs that should be measured among people with tendinopathy and establish whether currently available instruments are suitable and fit for purpose. Finally, future validation studies are required to determine how many people have clinically meaningful psychological factors in tendinopathy and identify appropriate cutoff levels for currently used instruments. Proposed cut-off values have been established for the FABQ [68] and TSK [55] for LBP, however, none currently exist in tendinopathy. Identifying individuals who may be at low, moderate or high risk of poor outcome would allow clinicians to adapt their management strategies accordingly.

## Conclusion

Synthesis of the cross-sectional data revealed low to very low certainty of evidence suggesting weak to moderate strength of association between psychological factors and pain, disability and physical functional outcome in tendinopathy. Importantly, data derived from cross-sectional studies reported similar strengths of association to those found in other common

musculoskeletal conditions, such as low back pain. Scarce and conflicting longitudinal data failed to show a predictive relationship between baseline psychological factors and long-term outcome. Hence, larger longitudinal cohort studies are required to investigate the predictive value of psychological factors upon long-term clinical outcome in tendinopathy populations, ensuring studies include people with meaningful levels of pain and psychological distress. Additionally, less heterogeneity of psychological outcome measures used in tendinopathy research is required; the development of a tendon specific measure may assist with this.

## Appendix 1

Database(s): Embase Classic+Embase 1947 to 2020 July 02

Search Strategy:

| # | Searches | Results |
|---|---|---|
| 1 | exp tendinitis/ | 18220 |
| 2 | tendinitis.tw. | 3270 |
| 3 | tend?nopath$.mp. [mp = title, abstract, heading word, drug trade name, original title, device manufacturer, drug manufacturer, device trade name, keyword, floating subheading word, candidate term word] | 5813 |
| 4 | exp achilles tendinitis/ | 1535 |
| 5 | achilles tend?n$.tw. | 11668 |
| 6 | paratenonitis.mp. [mp = title, abstract, heading word, drug trade name, original title, device manufacturer, drug manufacturer, device trade name, keyword, floating subheading word, candidate term word] | 82 |
| 7 | peritend?n$.mp. [mp = title, abstract, heading word, drug trade name, original title, device manufacturer, drug manufacturer, device trade name, keyword, floating subheading word, candidate term word] | 943 |
| 8 | jumpers knee.mp. [mp = title, abstract, heading word, drug trade name, original title, device manufacturer, drug manufacturer, device trade name, keyword, floating subheading word, candidate term word] | 327 |
| 9 | exp epicondylitis/ | 4048 |
| 10 | epicondyl$.tw. | 5600 |
| 11 | exp tennis elbow/ | 2966 |
| 12 | tennis elbow.tw. | 1282 |
| 13 | lateral epicondyl$.mp. [mp = title, abstract, heading word, drug trade name, original title, device manufacturer, drug manufacturer, device trade name, keyword, floating subheading word, candidate term word] | 2356 |
| 14 | exp shoulder impingement syndrome/ | 2830 |
| 15 | shoulder impingement syndrom$.tw. | 430 |
| 16 | exp rotator cuff injury/ | 11643 |
| 17 | rotator cuff injur$.tw. | 527 |
| 18 | rotator cuff disease.mp. [mp = title, abstract, heading word, drug trade name, original title, device manufacturer, drug manufacturer, device trade name, keyword, floating subheading word, candidate term word] | 602 |
| 19 | greater trochanteric pain syndrom$.mp. [mp = title, abstract, heading word, drug trade name, original title, device manufacturer, drug manufacturer, device trade name, keyword, floating subheading word, candidate term word] | 291 |
| 20 | exp fear/ | 274006 |
| 21 | fear$.tw. | 114290 |
| 22 | depression.tw. | 474458 |
| 23 | exp anxiety/ | 219742 |
| 24 | anxiety.tw. | 277800 |
| 25 | exp catastrophizing/ | 3552 |
| 26 | exp coping behavior/ | 64568 |

(*Continued*)

(Continued)

| # | Searches | Results |
|---|----------|---------|
| 27 | coping behavio$.tw. | 2499 |
| 28 | exp avoidance behavior/ | 36185 |
| 29 | avoidance behavi$.tw. | 5463 |
| 30 | exp avoidance behavior/ | 36185 |
| 31 | avoidance behavi$.tw. | 5463 |
| 32 | (fear adj1 avoidance).mp. [mp = title, abstract, heading word, drug trade name, original title, device manufacturer, drug manufacturer, device trade name, keyword, floating subheading word, candidate term word] | 1787 |
| 33 | (psycholog$ adj1 distress).mp. [mp = title, abstract, heading word, drug trade name, original title, device manufacturer, drug manufacturer, device trade name, keyword, floating subheading word, candidate term word] | 25744 |
| 34 | kinesiophobia.mp. [mp = title, abstract, heading word, drug trade name, original title, device manufacturer, drug manufacturer, device trade name, keyword, floating subheading word, candidate term word] | 1258 |
| 35 | hypervigilance.mp. [mp = title, abstract, heading word, drug trade name, original title, device manufacturer, drug manufacturer, device trade name, keyword, floating subheading word, candidate term word] | 1063 |
| 36 | catastrophi?ation.tw. | 195 |
| 37 | catastrophi?ing.mp. [mp = title, abstract, heading word, drug trade name, original title, device manufacturer, drug manufacturer, device trade name, keyword, floating subheading word, candidate term word] | 5704 |
| 38 | exp depression/ | 483960 |
| 39 | exp emotion/ | 619158 |
| 40 | emotion$.tw. | 271747 |
| 41 | psycholog$.mp. [mp = title, abstract, heading word, drug trade name, original title, device manufacturer, drug manufacturer, device trade name, keyword, floating subheading word, candidate term word] | 1097787 |
| 42 | tend?nosis.mp. [mp = title, abstract, heading word, drug trade name, original title, device manufacturer, drug manufacturer, device trade name, keyword, floating subheading word, candidate term word] | 1396 |
| 43 | subacromial pain.mp. [mp = title, abstract, heading word, drug trade name, original title, device manufacturer, drug manufacturer, device trade name, keyword, floating subheading word, candidate term word] | 160 |
| 44 | self efficacy.mp. | 33977 |
| 45 | exp plantar fasciitis/ | 1689 |
| 46 | plantar heel pain.mp. [mp = title, abstract, heading word, drug trade name, original title, device manufacturer, drug manufacturer, device trade name, keyword, floating subheading word, candidate term word] | 295 |
| 47 | exp physiological stress/ or exp stress/ | 337643 |
| 48 | 1 or 2 or 3 or 4 or 5 or 6 or 7 or 8 or 9 or 10 or 11 or 12 or 13 or 14 or 15 or 16 or 17 or 18 or 19 or 42 or 43 or 45 or 46 | 48811 |
| 49 | 20 or 21 or 22 or 23 or 24 or 25 or 26 or 27 or 28 or 29 or 30 or 31 or 32 or 33 or 34 or 35 or 36 or 37 or 38 or 39 or 40 or 41 or 44 or 47 | 2348181 |
| 50 | 48 and 49 | 1380 |

## Supporting information

**S1 Checklist. PRISMA 2009 checklist.**
(DOC)

## Author Contributions

**Conceptualization:** Carl Stubbs, Sean Mc Auliffe, Kieran O'sullivan, Peter Malliaras.

**Data curation:** Carl Stubbs, Sean Mc Auliffe, Adrian Mallows, Peter Malliaras.

**Methodology:** Carl Stubbs, Sean Mc Auliffe, Adrian Mallows, Kieran O'sullivan, Peter Malliaras.

**Project administration:** Carl Stubbs, Sean Mc Auliffe, Adrian Mallows, Kieran O'sullivan, Peter Malliaras.

**Supervision:** Sean Mc Auliffe, Adrian Mallows, Terence Haines, Peter Malliaras.

**Writing – original draft:** Carl Stubbs, Sean Mc Auliffe, Adrian Mallows, Kieran O'sullivan, Terence Haines, Peter Malliaras.

**Writing – review & editing:** Carl Stubbs, Sean Mc Auliffe, Adrian Mallows, Kieran O'sullivan, Terence Haines, Peter Malliaras.

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
