## [Decision Letter · Decision Letter 0]

24 Sep 2020

PONE-D-20-23274

The strength of association between psychological factors and clinical outcome

 in tendinopathy: a systematic review

PLOS ONE

Dear Dr. Mc Auliffe,

Thank you for submitting your manuscript to PLOS ONE. After careful consideration, we feel that it has merit but does not fully meet PLOS ONE’s publication criteria as it currently stands. Therefore, we invite you to submit a revised version of the manuscript that addresses the points raised during the review process.

Please see the reviewers' insightful recommendations for improving the quality of the manuscript.

We look forward to receiving your revised manuscript.

Kind regards,

Alison Rushton

Academic Editor

PLOS ONE

Reviewers' comments:

Reviewer's Responses to Questions

**Comments to the Author**

1. Is the manuscript technically sound, and do the data support the conclusions?

Reviewer #1: Yes

Reviewer #2: Yes

Reviewer #3: Yes

Reviewer #4: Yes

2. Has the statistical analysis been performed appropriately and rigorously? 

Reviewer #1: Yes

Reviewer #2: N/A

Reviewer #3: Yes

Reviewer #4: N/A

3. Have the authors made all data underlying the findings in their manuscript fully available?

Reviewer #1: Yes

Reviewer #2: No

Reviewer #3: Yes

Reviewer #4: Yes

4. Is the manuscript presented in an intelligible fashion and written in standard English?

Reviewer #1: Yes

Reviewer #2: Yes

Reviewer #3: Yes

Reviewer #4: Yes

5. Review Comments to the Author

Reviewer #1: 

The study tried to find an answer if there are strength of association between psychological factors and clinical outcomes in tendinopathy. It is a well written manuscript with rigorous literature search. However, there are minor observations:

1. Abstract- insert 'factors' after psychological. line 26

2. Introduction- 'MSK'. First mention. Write in full and then abbreviate (Line 64) and since you have abbreviated be consistent using it thereafter.

3. Methods- (SPADI, GROC). write full names. line 123.

4. Table 2- Format the column heading of table 2. indicate what represent SD and unit. e.g. Mean age±SD (year) and BMI± SD (unit). Delete SD in the rows.

5. References- check the journal for list of reference specifications. Punctuation and use of et al. for authors more than six were not consistent.

Reviewer #2: 

This systematic review summarized the current evidence on the association of psychological factors and clinical outcomes (disability and pain) in individuals with tendinopathy. The authors used accepted standards for conducting and reporting the review, including pre-registration, two reviewers for screening, extracting, and appraising. The manuscript is clear and easy to read and follow. The authors were limited in making the conclusions because of “very low” levels of certainty based on GRADE recommendations (also assessed by two authors). An additional limitation of this review is that the search terms used do not appear to be reflective of all tendinopathies in the body (see comments for methods section below) or of all psychological factors.

I have some comments below for improving the paper.

Major comments:

1. Provide detailed search terms used in at least one database with how search terms were modified for other databases. Search terms used are limited to a few types of tendinopathies and body regions. They do not reflect the intention of review to include all types of tendinopathies for all body regions. Similarly, the search terms do not reflect all psychological factors.

2. Data synthesis: Did the authors use any framework or guidelines to perform Quality synthesis? This is a factor that might have contributed to the Results section that is slightly tedious to read because of a lack of a priori framework for qualitative evidence synthesis.

3. It is unclear how was heterogeneity defined, and why meta-analysis was not performed. A priori criteria that authors used as criteria for meta-analysis are necessary.

4. Risk of Bias and overall GRADE rating: It is surprising to read that 9 of 10 studies had low risk of bias, however, based on GRADE assessment each category was classified as "very low levels of certainty" especially attributing to the risk of bias (80%). Am I missing something here?

Other comments:

Abstract:

1. Unclear what clinical outcomes were.

2. Stating how many authors performed screening, data extraction, and risk of bias assessment can be a good way to reflect the quality of the review through its Abstract.

3. Conclusions: new results should not appear in Conclusions: see reports on comparison with other musculoskeletal disorders and GRADE recommendations.

Methods:

1. Line 114 – I do not think that outcomes can be divided into Primary and Secondary for this type of review. These categories are better suited for reviews of RCTs.

2. SPADI, GROC - add full forms.

3. GROC is not a measure of disability.

4. Timing of outcome measures: Are there any reasons how the authors chose to categorise timing of outcomes into short, medium and long term? These are not consistent with Cochrane reviews. Please provide appropriate citations. Clarify if these were set a priori.

5. Line 148: unsure what the authors mean when they say ONE author "independently" screened the title and abstract. Independent of whom?

6. Line 160. It should be study design, not "type of trial" as observational studies were included.

7. Cite the papers included in Table 4 that informs the GRADE recommendations.

Results

1. Overall, the results could read better if the authors follow a qualitative synthesis guideline or a framework.

2. Lines 268-271- name what outcome measures were included.

Conclusions

Name what the clinical outcomes were to be unambiguous and explicit.

Reviewer #3: 

Thank you for a well-written and interesting manuscript! The length of the manuscript is exemplary. The research question is clear and the conclusion answers the question. A minor suggestion is to explain the abbreviation MSK in the introduction.

Reviewer #4: 

Thanks for giving me the opportunity to review this paper.

This is a very interesting systematic review paper reporting the magnitude of the association between psychological factors and clinical outcome in tendinopathy. The review identified ten studies that were assessed for quality using the modified Newcastle Ottawa Scale and certainty evidence by the GRADE Criteria. However there are some areas that the authors need to correct and adjust for more clarity as described below.

Title and abstract

1. The title is well written and captures all the necessary information from the study.

2. The abstract provides an informative and balanced summary of the appraisal. There is a need to provide the implication of the review for current and future practice.

3. It might not be requirement of the journal, but most of the time the registration number is provided at the end of the abstract.

Introduction

4. The introduction is well written explaining the rationale of the study.

5. It could be good to mention the setting where the study by Albers et al. was conducted (line 56-58). The same for the other studies in the same line.

6. I may suggest to mention the other treatment modalities used though exercises might have been shown to be effective in the management of tendinopathy (Line 59-66).

7. There is a need to elaborate more the problems related to the association focusing at methodological variability of the studies to support the idea of investigating the “strength of the association” (Line 76-78)

8. Review sentence (Line 76-78)

Methods and materials

9. The methods section is well written with enough details.

10. There is a need to mention what was based on to categorize the time of the outcome.

Results

11. The results are well presented with enough details

12. Study selection: Did not mention whether there are records identified through other sources as mentioned in the method section (Same observation for Figure 1)

13. The values presented in the results section should always match the findings from the reviewed studies.

a. Page 316: r=0.25 in text while in table it is r=0.27

b. Page 325: β=-0.41 in text while in table it is B 0.41

c. Page 344: r = -0.590 in text while in table it is r = ---590

14. Depending of where table 2 will be placed, the superscript reference number might change.

Discussion

15. The discussion summarises the main findings in the review and the strength of evidence for each main outcome

16. The example of musculoskeletal conditions used in the discussion is only low back pain and some other place rheumatological conditions. Can this mean that no other studies available that investigated the relationship between psychological factors and musculoskeletal pain or function.

17. What is the implication of the findings for practice recommendation?

Other comments

18. Line 9: “& These authors also contributed equally to this work” need review as there is only one author with “&” and it is better to mention the extent of the contribution.

19. Need consistency in referencing (check line 62, 73)

20. Corrections needed (Line 64)

21. Line 64: MSK should be written in full if first time used.

22. Insert reference (line 78-81)

23. Page 276: change “9-separate” into (nine separate)

24. Corrections needed (line 245, 317, 344, 424)

25. Line 353: “certatinty” into “certainty”

26. Line 359: “propsective” into “prospective”

27. Line 361: replace “our” with “the”

28. Line 363: “weak-to-moderate” into “weak to moderate”

29. Line 365: “50-cross-sectional” into “50 cross sectional”

6. PLOS authors have the option to publish the peer review history of their article (what does this mean?). If published, this will include your full peer review and any attached files.

Reviewer #1: No

Reviewer #2: **Yes: **Saurab Sharma, PhD

Reviewer #3: No

Reviewer #4: **Yes: **Assuman Nuhu

---

## [Author Response · Author response to Decision Letter 0]

16 Oct 2020

We thank the reviewers for their thoughtful and thorough review of our manuscript. We have taken every comment into consideration and responded to them individually in the response to reviewer document.

---

## [Decision Letter · Decision Letter 1]

5 Nov 2020

The strength of association between psychological factors and clinical outcome

 in tendinopathy: a systematic review

PONE-D-20-23274R1

Dear Dr. Mc Auliffe,

We’re pleased to inform you that your manuscript has been judged scientifically suitable for publication and will be formally accepted for publication once it meets all outstanding technical requirements.

Kind regards,

Alison Rushton

Academic Editor

PLOS ONE

Reviewers' comments:

Reviewer's Responses to Questions

**Comments to the Author**

1. If the authors have adequately addressed your comments raised in a previous round of review and you feel that this manuscript is now acceptable for publication, you may indicate that here to bypass the “Comments to the Author” section, enter your conflict of interest statement in the “Confidential to Editor” section, and submit your "Accept" recommendation.

Reviewer #1: All comments have been addressed

Reviewer #2: All comments have been addressed

Reviewer #4: All comments have been addressed

2. Is the manuscript technically sound, and do the data support the conclusions?

Reviewer #1: (No Response)

Reviewer #2: Yes

Reviewer #4: Yes

3. Has the statistical analysis been performed appropriately and rigorously? 

Reviewer #1: (No Response)

Reviewer #2: N/A

Reviewer #4: N/A

4. Have the authors made all data underlying the findings in their manuscript fully available?

Reviewer #1: (No Response)

Reviewer #2: Yes

Reviewer #4: Yes

5. Is the manuscript presented in an intelligible fashion and written in standard English?

Reviewer #1: (No Response)

Reviewer #2: Yes

Reviewer #4: Yes

6. Review Comments to the Author

Reviewer #1: (No Response)

Reviewer #2: I am happy with the responses provided and changes made in the revised manuscript. I am looking forward to read the paper in its published form. Best wishes!

Reviewer #4: Many thanks resending me the corrected manuscript.

All recommended revisions have been made. Acceptable for publication.

7. PLOS authors have the option to publish the peer review history of their article (what does this mean?). If published, this will include your full peer review and any attached files.

Reviewer #1: No

Reviewer #2: **Yes: **Saurab Sharma

Reviewer #4: No

---

## [Editor Report · Acceptance letter]

9 Nov 2020

PONE-D-20-23274R1 

The strength of association between psychological factors and clinical outcome
 in tendinopathy: a systematic review 

Dear Dr. Mc Auliffe:

I'm pleased to inform you that your manuscript has been deemed suitable for publication in PLOS ONE. Congratulations! Your manuscript is now with our production department. 

Kind regards, 

on behalf of

Professor Alison Rushton 

Academic Editor

PLOS ONE